# STRATEGIC DECEPTION IN DETERMINISTIC MARKOV DECISION PROCESSES VIA VALUE DIFFERENCES

## ABSTRACT

We investigate the design of autonomous deceptive agents capable of deceiving observers while executing tasks in deterministic and complex environments. Recent research has introduced an intent recognition model based on $Q$-differences and the *ambiguity model* (AM), which selects actions ambiguous over reward functions using pre-trained $Q$-functions to mislead observers. However, we identify that AM fails to achieve effective deception in deterministic Markov decision processes (DMDPs) because the strategy of maximizing entropy at each step leads to a large number of ineffective deceptive behaviors in the later stages of the task when the intention has been revealed. To address this problem, using the existing intent recognition based on state value differences ($V$-differences), we propose the concept of the last deceptive state (LDS), a method to compute the optimal LDS, and two $V$-differences-based Deceptive Models (VDMs). VDMs plan deceptive trajectories in DMDPs, moving beyond the geometric constraints of traditional path planning. Experiments in path planning domains demonstrate that VDMs achieve stronger deception and outperform AM across key metrics, including trajectory cost, deceptiveness, and steps after LDS.

## 1 INTRODUCTION

Deception, defined as the cultivation and sustained maintenance of false beliefs in others' minds (Carson, 2010), is a crucial skill that enables agents to gain strategic advantages by concealing their true intentions in adversarial environments. Its applications are widespread, spanning multi-agent negotiation (Greenberg, 1982; Matsubara & Yokoo, 1998), fugitive pursuit (Shieh et al., 2012), network intrusion (Geib & Goldman, 2001), business domain (Chelliah & Swamy, 2018) and privacy preservation (Keren et al., 2016). Non-player characters capable of deception are also known to create more credible and engaging interactions than those that behave honestly (Dias et al., 2013).

The challenge of executing a task while deceiving an observer is vividly illustrated in scenarios such as troop maneuver (Tzu, 2008). Here, a commander must move units to a target while avoiding ambush. Success hinges on balancing the efficiency of the movement with the need to conceal the final destination for as long as possible. Since moving directly toward the real target will inevitably reveal it, the troops must take deliberate, often circuitous, actions to mislead the adversary, thereby seizing the initiative. This work formulates and addresses this core problem within the framework of sequential decision-making under certainty.

A recent RL-based approach is the *Ambiguity Model* (AM) (Liu et al., 2021), using pre-trained Q-functions to estimate the probability of each candidate being real, and then chooses the action that maximizes the entropy of this distribution at each step. However, our analysis reveals a critical flaw in AM's application to deterministic Markov decision processes (DMDPs): Specifically, in Figure 1a and 1c, when an agent approaches the real goal state, the probability of the real goal state being identified becomes extremely high, where the agent's continued selection of maximum entropy actions is meaningless. The strategy of maximizing entropy at each step leads to a large number of redundant and ineffective deceptive behaviors in the later stages of the task when the intention has been revealed, which seriously affects efficiency. This deficiency motivates our key insight: it may be necessary for agents to explicitly reason about when to deceive and when to behave truthfully, making a clear distinction between the two.

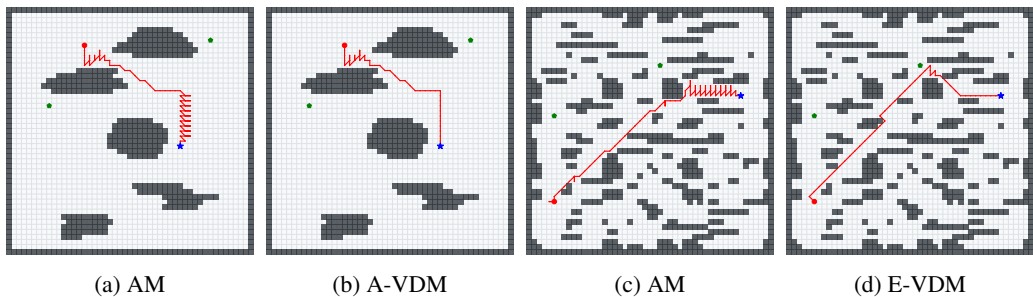

| (a) AM | (b) A-VDM | (c) AM | (d) E-VDM |

Figure 1: Deceptive trajectories generated by AM and our proposed VDMs. *Gray squares*: obstacles. *Red dot*: Agent start. *Blue five-pointed star*: Real goal state. *Green pentagons*: Fake goal states. *Red line*: Agent trajectory.

To address the limitations of the *Ambiguity Model* (AM) when applied to DMDPs, we draw inspiration from the concept of Last Deceptive Point (LDP) (Masters & Sardina, 2017b) and propose to identify a "critical state" that partitions the agent's plan into two distinct phases: a deceptive phase and a truthful phase. This division allows the agent to perform phase-specific planning, thereby achieving an effective balance between deception and efficiency.

However, we face the following challenges in our work: (1) How should this "critical state" be defined to meaningfully segment the planning process? (2) Does such a critical state exist for all DMDPs, and how can it be computed? (3) What planning strategies should the agent adopt in each phase to achieve effective deception?

To tackle these challenges, we propose *V*-differences-based Deceptive Models (VDMs) with three innovations: (1) We introduce the concept of the *Last Deceptive State* (LDS), which serves as an intermediate transitional state that clearly separates the agent's strategy into two phases—deceptive and truthful. This structure fundamentally differs from the blind entropy-maximizing behavior inherent in AM, enabling strategic deception in the first phase and direct optimal policy execution in the second, thereby avoiding inefficient pursuit of maximum entropy. Examples are illustrated in Figure 1b and 1d; (2) We prove the existence of the LDS in every DMDP and provide a method for finding the *optimal LDS*; (3) We propose a heuristic-based two-phase planning approach under the VDM framework, which achieves an optimal balance between deception potential and remaining efficiency. Specifically, VDMs mislead observers by either introducing ambiguity or exaggerating behaviors (Ambiguity-VDM and Exaggeration-VDM).

In summary, this work makes three contributions:

(1) We analyze the limitations of AM in solving problems of deceptive DMDPs;

(2) We propose the concept of the Last Deceptive State (LDS)—an intermediate transition state, a method to solve for the optimal LDS and *V*-differences-based Deceptive Models (VDMs), addressing the limitations in AM;

(3) We formalize deceptive planning within the DMDP framework using LDS. This generalization extends the problem beyond simple path-finding, allowing core concepts such as the LDS to apply to a broader class of sequential decision-making tasks.

## 2 RELATED WORK

Our work builds upon and connects ideas from several threads of research on computational deception, with a focus on planning in fully observable, deterministic environments.

The concept of deceptive path planning (DPP) was pioneered by Masters & Sardina (2017b), which introduced formal notions of simulation ("showing the false") and dissimulation ("hiding the real") (Bell, 2003; Whaley, 1982). A key idea in their work is the Last Deceptive Point (LDP), defined as the point in a path beyond which the real goal becomes the most likely, based on measuring deviation from optimal paths using cost differences. While LDP can be used to solve grid-based

deceptive path planning problems, it fails to address more general deceptive decision-making. We explore an analogous concept—the Last Deceptive State (LDS)—within the more general framework of Deterministic Markov Decision Processes (DMDPs). Furthermore, we utilize state value differences (*V*-differences) as a measure of progress, which accounts for expected long-term utility. The formalization of the LDS and the development of methods to compute it offer a value-based perspective on this planning problem, making our approach applicable to broader deceptive behavior decision-making scenarios beyond navigation.

Research has also extended deception to settings that account for uncertain environments. Ornik & Topcu (2018) incorporated the observer's belief into the agent's model, formalizing deceptive MDPs for stochastic environments. Similarly, Savas et al. (2022) proposed a *Deceptive Decision-Making (DDM)* algorithm that uses linear programming to generate trajectories which reach the real goal state with a certain probability under fixed time constraints, employing a maximum-entropy observer model, while we focus on deterministic environments and a rational observer model, which is predicated on the principle of maximum expected value (Ramírez & Geffner, 2010). Our work investigates a different, more constrained setting, aiming to enable the agent to stably reach the real goal state in deterministic environments.

Most closely related to our approach is the *Ambiguity Model (AM)* by Liu et al. (2021). AM models observers' beliefs via *Q*-differences-based plan recognition to measure the deviation between observed and optimal behavior and selects actions with maximum entropy at each step. Our analysis suggests that this approach may prolong deception unnecessarily after the agent's intent has become apparent. Lewis & Miller (2023) later introduced the *Deceptive Exploration Ambiguity Model (DEAM)* to address AM's limitations in model-free and continuous settings by integrating policy training with the deception objective, but it retains AM's core of selecting actions based on maximum entropy.

The deception strategies we have discussed are designed to counter the observer's capability of plan recognition, which is inherently the inverse problem of deception (Masters & Sardina, 2019). Plan recognition algorithms aim to infer an agent's goals and plans by analyzing its observed behavior. A widely adopted approach, and the one we assume our observer employs, is *cost-difference-based* recognition (Masters & Sardina, 2017a). This method quantifies how much an agent's behavior deviates from the optimal path to a potential goal; the smaller the cost differences, the more probable that goal is deemed to be. Our VDMs are designed specifically to mislead observers using such recognition models. By leveraging *V*-differences—a value-based generalization of cost-differences—VDMs effectively manipulate the observer's probability distribution over the goal state set $G$, thereby achieving strategic deception.

## 3 PRELIMINARIES

This section reviews the foundational concepts and models from prior work that are necessary to understand our proposed framework. The definition and observer model presented here are primarily based on the work of Savas et al. (2022), which we adapt and build upon for deterministic settings.

**Definition 1** (Deterministic Markov Decision Process). *A deterministic Markov decision process (DMDP) is a six-tuple $\mathcal{M} = (S, s_1, A, P, r, \gamma)$, where $s_1 \in S$ is the unique initial state; $A$ denotes a finite set of actions; $P : S \times A \times S \to \{0, 1\}$ is the deterministic state transition function: for any state $s \in S$ and action $a \in A$, there exists a unique successor state $s' \in S$ such that $P(s, a, s') = 1$, and the transition probability to all other states is 0 (i.e., executing action $a$ in state $s$ necessarily transitions to $s'$); $r : S \times A \times S \to \mathbb{R}$ is the reward received when executing action $a$ in state $s$ and transitioning to state $s'$; $\gamma \in [0, 1]$ is the discount factor.*

Problems related to DMDPs are solved by finding a policy $\pi : S \to A$, which prescribes the action an agent should take in a given state. The objective is to maximize the value function, defined as: $V_\pi(s) = \mathbb{E}\left[\sum_{t=0}^{\infty} \gamma^t r(s_t, a_t, s_{t+1})\right]$.

A trajectory $\zeta$ is a sequence of states and actions $(s_1, a_1, s_2, a_2, s_3, \dots)$ such that $P(s_t, a_t, s_{t+1}) = 1$ for all $t \in \mathbb{N}$. A partial trajectory $\zeta_{1:t}$ of length $t \in \mathbb{N}$ is the sequence $(s_1, a_1, s_2, \dots, s_t)$. Let $\zeta_{1:t}[n] = s_n$ with $1 \leq n \leq t$ denote the state visited at the $n$-th step along $\zeta_{1:t}$.

Savas et al. (2022) proposed a predictive model that integrates the reasoning method from the observer's maximum-entropy reinforcement learning model with rationality assumptions, computing a probability $\Pr(g \mid \zeta_{1:t})$ for each potential goal $g \in G$. Prior beliefs form a probability distribution over the agent's potential goals, formalizing the observer's judgment of "the goal state the agent is expected to reach from the initial state $s_1$". Priors are typically uniform; in repeated or specialized interactions, they can be constructed from historical data via Bayesian methods (Ziebart et al., 2009). Given the prior, the observer's posterior prediction of the goal follows Bayes' theorem:

$$\Pr(g \mid \zeta_{1:t}) = \frac{\Pr(\zeta_{1:t} \mid g)\Pr(g)}{\sum_{g' \in G}\Pr(\zeta_{1:t} \mid g')\Pr(g')},$$

where $\Pr(\zeta_{1:t} \mid g)$ denotes the probability that the agent follows trajectory $\zeta_{1:t}$ to reach $g$ (i.e., the observer's expectation of "how the agent reaches the goal").

Savas et al. (2022)'s intent recognition model provides a formula for calculating probabilities of goal states based on $V$-differences. When transition randomness has a limited impact on the agent's behavior and the discount factor $\gamma$ is sufficiently large, for the softmax entropy iteration $V_g(s) = \text{softmax}_{a \in A} Q_g(s, a)$, we have:

$$\Pr(g \mid \zeta_{1:t}) \approx \frac{\exp(\Delta V_g(\zeta_{1:t}))\Pr(g)}{\sum_{g' \in G}\exp(\Delta V_{g'}(\zeta_{1:t}))\Pr(g')},$$

where $\text{softmax}_\alpha f(x) = \alpha \log \sum_{g' \in G}\exp(f(x)/\alpha)$ and $\Delta V_g(\zeta_{1:t}) = \Delta V_g(s_1, s_t) = V_g(s_t) - V_g(s_1)$. Intuitively, $\Delta V_g(\zeta_{1:t})$ represents the state value growth for goal state $g$ after the agent follows trajectory $\zeta_{1:t}$. DDM predicts the observer's model by controlling $\alpha$: as $\alpha \to \infty$, the observer deems the trajectory inefficient and assumes random behavior (equal posterior probabilities for all goal states); as $\alpha \to 0$, and it is predicted that the agent is fully efficient, following the least-cost trajectory (Savas et al., 2022).

==The probability calculation based on *V*-differences is well-suited for DMDPs given their deterministic transitions.== This approach leverages the disparities between $V$-functions of states to quantify the potential value gain of the agent transitioning from the initial state $s_1$ to the current state $s_t$.

## 4 DECEPTIVE TRAJECTORY PLANNING

### 4.1 MODELING OBSERVER PREDICTIONS

To model the observer's inference, we adopt the value-difference-based recognition model introduced by Savas et al. (2022). However, building on the intuition that a rational agent is most likely to follow an optimal (minimum-cost) or near-optimal goal plan Ramírez & Geffner (2010), a key distinction in our work is the assumption of a perfectly rational observer ($\alpha \to 0$). This reduces the value iteration for $V_g(s)$ to $V_g(s) = \arg\max_{a \in A} Q_g(s, a)$. We assume uniform prior probabilities and derive the posterior probability formula for goal states based on $V$-differences:

$$\Pr(g \mid \zeta_{1:t}) \approx \frac{\exp(\Delta V_g(\zeta_{1:t}))}{\sum_{g' \in G}\exp(\Delta V_{g'}(\zeta_{1:t}))}.$$

Thus, the observer's prediction $\Pr(g \mid \zeta_{1:t})$ is exclusively dependent on the agent's initial state $s_1$ and current state $s_t$, i.e., $\Pr(g \mid \zeta_{1:t}) = \Pr(g \mid s_1, s_t)$, which aligns with the memoryless property of Markov decision processes. We proceed to derive the observer's prediction model from the agent's expected policy Sutton et al. (1998). The observer's probability prediction for each goal state is obtained via offline computation of $V_g(s)$ for all $g \in G$ and $s \in S$ using value iteration. ==We assume that all goal states are absorbing, i.e., $P(g, a, g) = 1$ for all $g \in G$.== The optimal value function $V_g^*(s)$ is computed iteratively until convergence, satisfying the Bellman optimality equation:

$$Q_g(s, a) = -\text{cost}(s, a) + \gamma \sum_{s' \in S} P(s, a, s')V_g(s'), \quad V_g(s) = \arg\max_{a \in A} Q_g(s, a)$$

The values of $V_g(s)$ and $Q_g(s, a)$ can be iteratively computed via value iteration using the initialization $V_g(g) = M$ and $V_g(s) = 0$ for all $s \in S \setminus \{G\}$, where $M$ is an arbitrarily large constant (Baier & Katoen, 2008). The optimal policy can be derived as $\pi_g^*(s) = \arg\max_{a \in A} Q_g(s, a)$.

### 4.2 COMPONENTS OF A DECEPTIVE TRAJECTORY

To further structure our deceptive planning framework, we first introduce several key concepts that characterize the agent's state and actions from the perspective of the observer's beliefs. These components allow us to formally define the critical juncture—the Last Deceptive State—where the agent's strategy should transition from deception to truthfulness.

**Definition 2** (Truthful State). *A **truthful state** is a state $s \in S$ at which the probability of the real goal state $g_r$ is greater than the probability of any other goal states in $G$. Formally, for all $g \in G \setminus \{g_r\}$, $Pr(g_r \mid s_1, s) > Pr(g \mid s_1, s)$. Otherwise, this state is a **deceptive state**.*

The binary classification of states is fundamental. It allows the agent (and our planning algorithm) to know, at any point, whether it is currently successfully deceiving the observer ($s$ is deceptive) or if its true intent has been uncovered ($s$ is truthful).

**Definition 3** (Truthful Action). *A **truthful action** is an action $a \in A(s)$ if the state $s'$ is a truthful state, where $s' = T(s, a)$ denotes the state transition from state $s$ via action $a$. Otherwise, $a$ is a **deceptive action**.*

This definition connects actions to the deception strategy. Selecting a deceptive action is the means by which the agent maintains ambiguity, while taking a truthful action implies the agent has chosen to (or must) reveal its intent.

**Definition 4** (Last Deceptive State (LDS)). *For a trajectory $\zeta_{1:t}$, its **last deceptive state** is a state $s \in S$ satisfying:*

- *$\zeta_{1:t}[t] = s_t = s$ is a deceptive state;*

- *For all $j \in \{t+1, \ldots, T\}$ and $g \in G \setminus \{g_r\}$, $Pr(g_r \mid \zeta_{1:j}) \geq Pr(g \mid \zeta_{1:j})$.*

The LDS is the cornerstone of our two-phase planning paradigm. It defines the precise state where deception ends and optimal progress toward the real goal state begins. After the LDS, all subsequent states are truthful (by Definition 2), meaning the observer is no longer deceived, and thus the agent should simply follow the optimal policy to $g_r$. Identifying this state is therefore the key to balancing deception and efficiency.

**Corollary 1.** *The LDS always exists, because the initial state $s_1$ has uniform prior probabilities over all goal states, i.e., $Pr(g_r \mid \zeta_{1:1}) = Pr(g \mid \zeta_{1:1})$ for all $g \in G \setminus \{g_r\}$, thus $s_1$ is inherently a deceptive state.*

**Note:** This corollary ensures that our search for an LDS is well-founded, as every trajectory begins under a state of ambiguity.

A fully observable deceptive trajectory eventually becomes "truthful" as its final action necessarily leads to the real goal state $g_r$. Thus, for an agent with multiple goal states, every trajectory contains a deceptive state such that all subsequent states are truthful.

### 4.3 MAXIMIZING THE VALUE OF LAST DECEPTIVE STATE

Knowing that an LDS exists is not sufficient for planning; we need to find the best one. This section formalizes the properties of deceptive states to ultimately derive a method for computing the optimal LDS, which will serve as the anchor point for our VDMs in Section 5.

**Definition 5** (Deceptive States Set $D$). *For an agent with initial state $s_1$, the **deceptive states set** $D$ contains all states $s \in S$ where there exists $g \in G \setminus \{g_r\}$ such that $Pr(g_r \mid s_1, s) \leq Pr(g \mid s_1, s)$ (i.e., $\Delta V_g(s_1, s) \geq \Delta V_{g_r}(s_1, s)$).*

This set $D$ is our entire search space for potential LDS candidates. It contains every state from which the observer is not yet certain of the real goal state.

However, not all states in $D$ are viable candidates for the LDS. We now identify subsets of $D$ that have specific properties related to how the agent can leave the set of deceptive states.

**Definition 6** (Edge Deceptive States Set $D^e$). *The **edge deceptive states set** $D^e \subseteq D$ is defined such that for any $s \in D^e$, if there exists an action $a \in A(s)$ where $V_{g_r}(s') > V_{g_r}(s)$ (with $s' = T(s, a)$), then $s' \notin D$.*

**Intuition:** A state is on the "edge" of the deceptive set if there *exists* an action that improves the agent's value with respect to the real goal state *immediately forces it to exit* the deceptive set (i.e., the next state becomes truthful). This is a critical property for the LDS.

**Definition 7** (Critical Deceptive States Set $D^c$). *The **critical deceptive states set** $D^c \subseteq D^e$ is defined such that for any $s \in D^c$ and all $a \in A(s)$, if $V_{g_r}(s') > V_{g_r}(s)$, then $s' \notin D$.*

**Intuition:** This is a stricter subset of the edge states. For a critical state, *it is guaranteed* that any action that improves the agent's position toward $g_r$ will make the next state truthful. This makes critical states strong candidates for the LDS.

**Theorem 1.** *For any $s \in D$ that can reach $g_r$ (i.e., there exists a trajectory from $s$ to $g_r$) , there exists an action $a \in A(s)$ such that $V_{g_r}(s') > V_{g_r}(s)$.*

*Proof.* Since $s \in D$, $s \neq g_r$ (otherwise $s$ would be a truthful state). Since an agent can always reach $g_r$ from state $s$, there must exist an action $a \in A(s)$ leading to some $s'$ where $V_{g_r}(s') > V_{g_r}(s)$ (otherwise $V_{g_r}(g_r)$ could not be achieved). Thus, $V_{g_r}(g_r) \geq V_{g_r}(s') > V_{g_r}(s)$. $\qquad\square$

**Theorem 2.** *For any $s \in D \setminus D^e$ and all $a \in A(s)$: If $V_{g_r}(s') > V_{g_r}(s)$, then $s' \in D$.*

*Proof.* Assume for contradiction that $s' \notin D$. By Definition 6, if all actions $a \in A(s)$ leading to some $s'$ where $V_{g_r}(s') > V_{g_r}(s)$ and $s' \notin D$, then $s \in D^e$. This contradicts $s \in D \setminus D^e$, so $s'$ must belong to $D$. $\qquad\square$

**Corollary 2.** *No state $s \in D \setminus D^e$ can be the Last Deceptive State (LDS).*

**Implication:** Theorem 2 and Corollary 2 are crucial as they tell us that the LDS must be found within the set of edge deceptive states $D^e$. Any non-edge state has a path to stay inside $D$ while still making progress toward $g_r$, meaning deception can and should continue. The LDS is the point where this is no longer possible.

**Corollary 3.** *For any state $s \in D \setminus D^e$, there exists an action $a \in A(s)$ such that $V_{g_r}(s') > V_{g_r}(s)$ and $s' \in D$.*

**Theorem 3.** *Given a state $s^* \in D$ and $V_{g_r}(s^*) \geq V_{g_r}(s)$ for all $s \in D$, then $s^* \in D^c$.*

*Proof.* Suppose there exists an action $a \in A(s^*)$ such that $V_{g_r}(T(s^*, a)) > V_{g_r}(s^*)$, which contradicts the given condition that $V_{g_r}(s^*) \geq V_{g_r}(s)$ for all $s \in D$ if $T(s^*, a) \in D$. The assumption is invalid, and such an action $a$ does not exist. According to Definition 7, $s^* \in D^c$. $\qquad\square$

**Implication:** Theorem 3 is the important. It states that the deceptive state with the highest value (i.e., the one closest to $g_r$ in terms of remaining cost/effort) is necessarily a critical state. This provides a straightforward and computable method for finding the optimal LDS: it is the state in $D$ that maximizes $V_{g_r}(s)$. Conceptually, this state offers the best possible trade-off: it is the farthest into the deceptive phase that the agent can go while still ensuring that the subsequent truthful phase (the trajectory from $d_c^*$ to $g_r$) is as short and efficient as possible.

This critical deceptive state, $d_c^* = \arg\max_{s \in D} V_{g_r}(s)$, is the **optimal Last Deceptive State**. It is the linchpin of our VDMs (introduced in Section 5), as it defines the transition point between the deceptive and truthful phases of the planned trajectory.

We also identify that the optimal decoy goal state is $g^* = \arg\max_{g \in G \setminus \{g_r\}} [V_g(d_c^*) - V_g(s_1)]$ according to $d_c^*$. Since $\Delta V_g(s_1, d_c^*)$ directly determines the posterior probability $\Pr(g \mid \zeta_{1:t}, \zeta_{1:t}[t] = d_c^*)$, $g^*$ represents the decoy goal state with the maximum posterior probability given the agent's state $d_c^*$, making it the decoy goal state most likely to be mistaken for the true one. This information is utilized by our Exaggeration-VDM model to maximize deception during the planning phase.

## 5 V-DIFFERENCES-BASED DECEPTIVE MODELS

Building upon the foundational concepts of $d_c^*$ and $g^*$ introduced above, we present *V*-differences-based Deceptive Models (VDMs). VDMs exhibit three key characteristics: (1) They introduce two stages for the entire deceptive planning, clarifying the strategies that the agent should adopt in each

stage; (2) They propose two planning models based on ambiguity ("hiding the real") and exaggeration ("showing the false") respectively and a deception factor in the exaggeration model, enabling the overall adjustability of deception and cost and thus enhancing its adaptability to different environments; (3) Their training efficiency overhead compare to AM is $\frac{|G|+1}{|G|}$. Both leverage value iteration for precomputing state values: AM requires $|G|$ iterations (one per candidate goal state in $G$), while VDMs add only one more iteration targeting $d_c^*$, resulting in $|G|+1$ total iterations.

Through a two-phase planning paradigm centered on the critical deceptive state $d_c^*$, VDMs decompose trajectory planning into two distinct phases:

$$\zeta = \underbrace{s_1 \xrightarrow{\text{deceptive}} d_c^*}_{\text{Stage 1}} \oplus \underbrace{d_c^* \xrightarrow{\text{optimal}} g_r}_{\text{Stage 2}}.$$

**Stage 1 (Deceptive Phase):** Transition from initial state $s_1$ to the optimal LDS $d_c^*$ while pursuing deception. This phase leverages either ambiguity or exaggeration strategies.

**Stage 2 (Truthful Phase):** Execute optimal policy from $d_c^*$ to $g_r$. Since all post-$d_c^*$ states are truthful (Definition 4), deception is impossible, and efficiency is prioritized.

The separation ensures: (1) Maximum feasible deception in Stage 1; (2) Guaranteed the real goal state achievement in Stage 2. We now instantiate this framework with two concrete models.

**Ambiguity-VDM:** Extending AM's principle, *Ambiguity-VDM* implements dissimulation during Stage 1 through entropy-maximizing actions, using $V$-differences to calculate probabilities and select action rather than $Q$-differences:

$$a^* = \underset{a \in A(s)}{\arg\max} \, H\left(\Pr(g \mid (s_1, s) \cup s')\right),$$

where $s$ is the current state, $s' = T(s, a)$, $H(\cdot)$ is Shannon entropy (Shannon, 1948) and probabilities are computed via:

$$\Pr(g \mid s_1, s') = \frac{\exp\left(V_g(s') - V_g(s_1)\right)}{\sum_{g' \in G} \exp\left(V_{g'}(s') - V_{g'}(s_1)\right)}.$$

Compared with AM, the *Ambiguity-VDM* constrains ambiguous behavior to Stage 1, uses $d_c^*$ as the deception horizon to prevent over-pursuit of maximum entropy, and maintains trajectory efficiency via Stage 2 optimization. A performance comparison is shown in Figure 1a and 1b.

**Exaggeration-VDM:** Inspired by DPP (Masters & Sardina, 2017b), we propose two VDMs about exaggeration:

**Parameterized:** *Parameterized Exaggeration-VDM* strategically lures observers toward $d_c^*$ with adjustable deception level. This model implements:

1. *Pruning*: Discard actions that do not increase $V_{d_c^*}(s)$:
$$A^+(s) = \left\{a \in A(s) \mid V_{d_c^*}(s') > V_{d_c^*}(s)\right\}.$$

2. *Action selection*: From remaining actions, maximize:
$$a^* = \underset{a \in A^+(s)}{\arg\max}\left[\Delta V_{d_c^*}(s, s') + \sigma \cdot \left(\Delta V_{g^*}(s, s') - \Delta V_{g_r}(s, s')\right)\right].$$

Pruning is to ensure that the state of the agent in Stage 1 will necessarily converge to $d_c^*$. $\sigma \geq 0$ is a deception weight. Higher $\sigma$ enhances deception at trajectory cost. The term $\Delta V_{g^*}(\cdot) - \Delta V_{g_r}(\cdot)$ quantifies deception effectiveness by measuring how actions increase the relative value of the decoy and real goal state.

**Decoy-First:** *Decoy-First Exaggeration-VDM* implements a navigation to the optimal decoy goal state $g^*$ during Stage 1:

$$\zeta = \underbrace{s_1 \xrightarrow{\text{optimal}} g^* \oplus g^* \xrightarrow{\text{optimal}} d_c^*}_{\text{Stage 1}} \oplus \underbrace{d_c^* \xrightarrow{\text{optimal}} g_r}_{\text{Stage 2}}.$$

The model explicitly reaches the optimal decoy goal state $g^*$ before the optimal LDS $d_c^*$, creating a strong deception signal ("showing the false").

For conciseness in subsequent sections, we refer to the *Ambiguity-VDM* as "A-VDM", the *Exaggeration-VDM(Parameterized)* as "E-VDM($\sigma = c$)", where $c \geq 0$ and the *Exaggeration-VDM(Decoy-First)* as "E-VDM$^*$". In other words, the prefixes "E" and "A" denote "exaggeration" and "ambiguity" respectively.

## 6 EVALUATION

We conducted the experiments using observers to estimate the probability distribution over candidate goal states, following the experimental setup by Liu et al. (2021). Our experiments aim to evaluate whether VDMs address the limitations of AM in DMDPs, and compare the performance and efficiency of the exaggerated and ambiguous trajectories with two other algorithms. We compare the following methods: (1) Honest (moves directly to the real goal state); (2) AM; (3) A-DDM($\gamma_a = 0.95$); (4) E-DDM($\gamma_a = 0.95$); (5) A-VDM; (6) E-VDM($\sigma = 1$); (7) E-VDM$^*$. The parameter $\gamma_a$ in DDM was selected in accordance with Savas et al. (2022). All of the comparative models were evaluated under the same observer model. All experiments were conducted on a machine with a 3.8GHz CPU and 24GB RAM, and we employ the Gurobi 12.0.3 solver for DDM's linear programming.

### 6.1 EXPERIMENT DESIGN

**Environment**: We evaluate agents in a discrete path planning environment, where the agent can move laterally or diagonally to reach the real goal state. The agent receives a reward of 100 for reaching the goal states, $-1$ for lateral actions, and $-\sqrt{2}$ for diagonal actions. We set $\gamma = 0.99$ and use 40 environmental setups (5 maps × 8 goal states) consistent with those in Liu et al. (2021). These maps vary in size and obstacle configurations as follows: (1) 49×49 with no obstacles; (2) 49×49 with several large obstacles; (3) 49×49 with many small obstacles; (4) 100×100 with large "island" obstacles; (5) 100×100 maze-type maps.

**Metrics**: We use the following metrics: (1) *Trajectory Cost*: The ratio of the trajectory cost generated by different models to that of the Honest model. It is used to measure the cost efficiency of trajectories; (2) *Deceptiveness*: Following Masters & Sardina (2017b), deceptiveness is measured by the probability of achieving the real goal state at different percentages of trajectory completion; (3) *Steps after LDS*: According to Definition 4, a rational observer will not be deceived after the LDS from their perspective. Thus, fewer steps after the LDS indicate better deception. (4) *Generation Time*: The time required for each model to generate a complete deceptive trajectory.

### 6.2 EXPERIMENT RESULTS

**Trajectory Cost**. Figure 2a shows that E-VDM($\sigma = 1$) and E-VDM$^*$ are more cost-efficient than A-VDM. The AM and A-VDM strategies force the agent to select entropy-maximizing ambiguous actions, rather than planning decisively toward relevant states, leading to higher trajectory costs. A-VDM has lower trajectory costs than AM because we decompose planning into two stages and ensure efficiency in Stage 2 of trajectories. This restricts ambiguous decision-making to the intermediate state $d_c^*$, reducing costs.

When comparing VDMs with DDM (Figure 2d), the Kruskal-Wallis test revealed significant differences across methods ($p < 0.0001$). Paired $t$-tests showed that A-DDM($\gamma_a = 0.95$) achieved the lowest trajectory cost ratio (mean = 1.07), significantly outperforming A-VDM (mean = 1.78, $p < 0.0001$). However, E-VDM($\sigma = 1$) (mean = 1.35) demonstrated substantially better cost efficiency than E-DDM($\gamma_a = 0.95$) (mean = 2.03, $p < 0.0001$), representing a 33% reduction in trajectory cost. This indicates that while A-DDM achieves high efficiency through ambiguity-based strategies, E-VDM provides a more balanced trade-off between deception and efficiency in exaggeration-based planning.

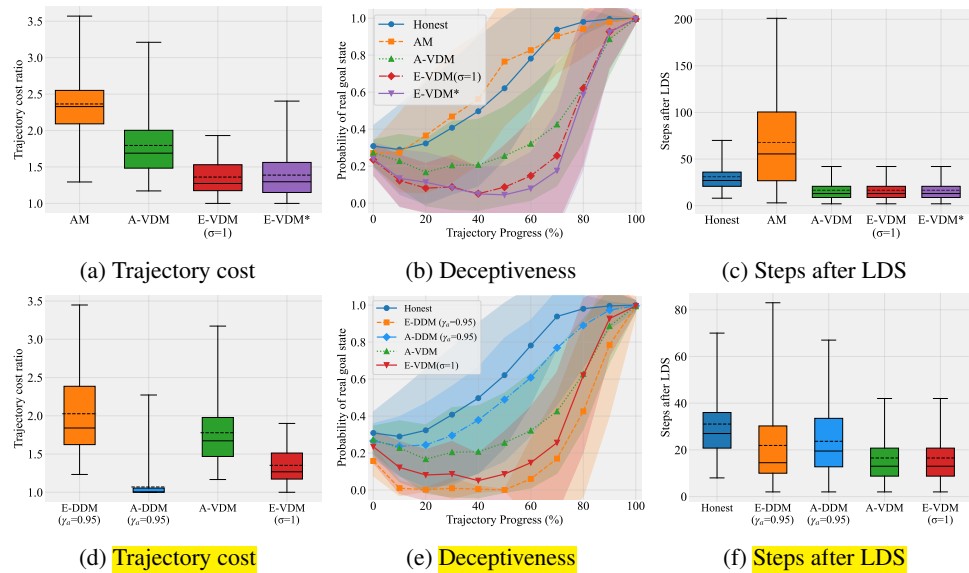

(a) Trajectory cost      (b) Deceptiveness      (c) Steps after LDS

(d) Trajectory cost      (e) Deceptiveness      (f) Steps after LDS

Figure 2: Shaded regions represent the standard deviation (SD) around the mean for each model. In the box plots, solid lines represent medians, dashed lines represent means, and shaded areas represent interquartile ranges.

**Deceptiveness**. Figure 2b illustrates how the probability of the real goal state $g_r$ evolves as the agent progresses along the trajectories. Since the agent eventually moves toward and reaches $g_r$, the probability increases over time. The Kruskal-Wallis test revealed significant differences in probability distributions of $g_r$ across models ($p < 0.0001$). One-tailed paired $t$-tests for relative deceptiveness showed that A-VDM ($p < 0.0001$) and E-VDM($\sigma = 1$) ($p < 0.0001$) had significantly lower average probabilities than Honest and AM. AM's average probability was not significantly lower than Honest ($p \approx 0.83$). Additionally, E-VDM($\sigma = 1$) had a significantly lower average probability than A-VDM ($p < 0.0001$), demonstrating that VDMs effectively achieve deception. There is no significant difference between E-VDM($\sigma = 1$) and E-VDM* ($p \approx 0.07$).

Comparing with DDM methods (Figure 2e, E-DDM($\gamma_a = 0.95$) achieved the strongest deceptiveness (mean probability = 0.21), significantly outperforming all other methods including E-VDM($\sigma = 1$) (mean = 0.31, $p < 0.0001$). Notably, A-VDM (mean = 0.41) demonstrated stronger deceptiveness than A-DDM($\gamma_a = 0.95$) (mean = 0.56, $p < 0.0001$), showing that the V-differences-based ambiguity strategy in VDMs is more effective than DDM's ambiguity approach.

**Steps after LDS**. Figure 2c shows that A-VDM, E-VDM($\sigma = 1$), and E-VDM* exhibit identical distributions, as all three use $d_c^*$ as an intermediate transition state. The Honest model produces slightly longer steps after LDS than VDMs. In contrast, AM's excessive pursuit of entropy-maximizing ambiguous actions causes unnecessary delays, leading to significantly more steps after LDS.

The comparison with DDM methods (Figure 2f) demonstrates a key advantage of VDMs. The Kruskal-Wallis test confirmed significant differences across methods ($p < 0.0001$). Both A-VDM and E-VDM($\sigma = 1$) (mean = 16.5 steps) significantly outperformed their DDM counterparts: A-VDM vs. A-DDM($\gamma_a = 0.95$) (mean = 23.7, $p < 0.0001$) and E-VDM($\sigma = 1$) vs. E-DDM($\gamma_a = 0.95$) (mean = 21.9, $p < 0.01$). This reduction in steps after LDS indicates that VDMs' explicit reasoning about the last deceptive state through $d_c^*$ shortens the duration of the real goal state's sustained exposure in the later task phase.

**Generation Time**. Table 1 presents the computational time required to generate complete deceptive trajectories across different map sizes. All methods include $|G|$ value iterations preprocessing time for candidate goals, ensuring a fair comparison of total generation time. For 49×49 maps, Honest and AM require minimal time (mean ≈ 8.5s), as both only need $|G|$ value iterations for candidate goals plus negligible planning overhead. VDMs (A-VDM and E-VDM) require moderately more

Table 1: Computational time comparison (seconds) across map sizes. All methods include $|G|$ value iterations preprocessing time for candidate goals. Values are mean $\pm$ std over 24 problems (49$\times$49) or 16 problems (100$\times$100).

| Map Size | Honest | AM | A-VDM | E-VDM $(\sigma = 1)$ | A-DDM $(\gamma_a = 0.95)$ | E-DDM $(\gamma_a = 0.95)$ |
|---|---|---|---|---|---|---|
| 49$\times$49 | 8.5$\pm$2.9 | 8.5$\pm$2.9 | 11.0$\pm$3.3 | 11.0$\pm$3.3 | 26.1$\pm$5.0 | 26.1$\pm$4.6 |
| 100$\times$100 | 71.6$\pm$17.2 | 71.3$\pm$17.1 | 91.8$\pm$15.0 | 91.7$\pm$15.0 | 777.3$\pm$96.7 | 772.6$\pm$109.2 |

time (mean $\approx$ 11.0s), representing a 29% increase over AM due to one additional value iteration for computing $d_c^*$. In contrast, DDM methods require substantially more time: A-DDM($\gamma_a = 0.95$) and E-DDM($\gamma_a = 0.95$) average 26.1s, representing 137% additional overhead compared to VDMs due to solving linear programs at each trajectory step.

For 100$\times$100 maps, this computational gap becomes even more pronounced. Honest and AM require 71.3–71.6s, VDMs require 91.7–91.8s (29% increase over AM), while DDM methods require 772.6–777.3s—representing an order of magnitude slowdowns compared to VDMs. The minimal overhead of VDMs over AM (only one additional value iteration for $d_c^*$) makes them practical for real-world deployment where both deception and efficiency are critical.

### 6.3 HYPERPARAMETER STUDIES FOR E-VDM($\sigma = c$)

E-VDM($\sigma = c$) introduces the hyperparameter $\sigma$. Figure 3a–3c present results regarding deceptiveness and trajectory cost. A larger deception weight $\sigma$ yields a lower real goal state probability and a higher trajectory cost (reported as ratios relative to those of E-VDM($\sigma = 0$)), indicating that $\sigma$ balances the efficiency and deceptiveness.

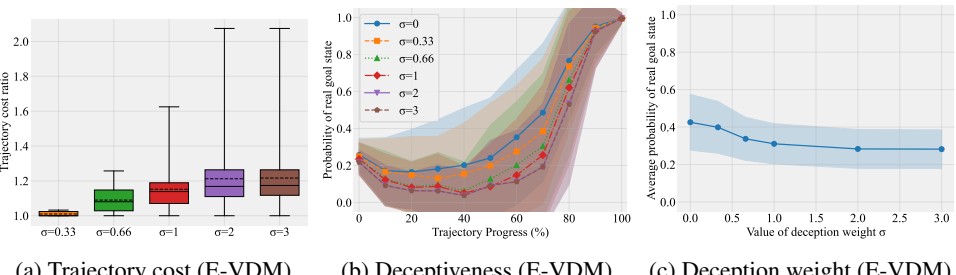

(a) Trajectory cost (E-VDM)  (b) Deceptiveness (E-VDM)  (c) Deception weight (E-VDM)

Figure 3: E-VDM's sensitivity to hyperparameters: deception weigh $\sigma$.

As $\sigma \to 0$, E-VDM($\sigma = c$) exhibits a propensity to converge rapidly to $d_c^*$, with a prioritization of efficiency over deceptiveness. Specifically, one-tailed paired $t$-tests revealed no statistically significant difference between E-VDM($\sigma = 0$) and A-VDM. As $\sigma$ increases, the trajectory cost of E-VDM rises, while the average probability of the real goal state decreases, with this trend decelerating. The deceptiveness at $\sigma = 2$ is statistically significantly higher than that at $\sigma = 1$; however, the improvements in both deceptiveness and trajectory cost from $\sigma = 2$ to $\sigma = 3$ are negligible. The "Steps after LDS" metric appears to be unaffected by $\sigma$, and thus, no further analysis is provided for this metric.

## 7 CONCLUSION

We have presented *V*-Differences-based Deceptive Models (VDMs), a novel deceptive planning framework for deterministic Markov decision processes (DMDPs). The framework introduces the Last Deceptive State (LDS)—whose existence we prove for DMDPs—along with a method to compute the optimal LDS. We further propose two deception strategies based on ambiguity and exaggeration. Experiments confirm that VDMs outperform AM in DMDPs. Crucially, our work generalizes the Last Deceptive Point (LDP) concept to a broader class of sequential decision-making tasks.

This work has several limitations. First, our proposed VDMs are designed for DMDPs and may not be suitable when transition stochasticity significantly influences the agent's behavior, so that VDMs have not completely solved the problem of AM. Second, we assume a fully rational observer. Third, our environment is assumed to be fully observable. Finally, our evaluation is conducted solely in path-planning settings. In future work, we plan to incorporate more sophisticated observer models and extend VDMs to more general classes of Markov decision processes (MDPs).

**Reproducibility Statement**   To facilitate the reproducibility of our work, we have taken the following measures: (1) For the theoretical contributions in Sections 3 and 4, including the definition of the Last Deceptive State (LDS) and the proposed VDMs, we provide complete proofs and detailed algorithmic descriptions in the main text. (2) For the experimental results in Section 6, we include a comprehensive description of the experimental setup, including environment specifications, observer models, and evaluation metrics. (3) The source code implementing our models and experiments, has been included as part of the supplementary materials submitted anonymously. (4) To aid reproducibility, we have also preserved selected experimental results and the trained model files (.npz format) in the supplementary materials, providing additional resources for understanding and verifying our findings. We encourage readers to refer to the respective sections and supplementary materials for further details.

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

# A APPENDIX

## LARGE LANGUAGE MODEL USAGE STATEMENT

In accordance with ICLR 2026 policy on the use of large language models (LLMs), we provide the following transparent disclosure regarding the role of LLMs in the preparation of this manuscript.

The authors utilized DeepSeek-V3, a large language model developed by DeepSeek, exclusively as an assistive tool for improving the presentation quality of this paper. Specifically, the model was employed to:

1. Enhance the clarity, coherence, and grammatical accuracy of certain paragraphs;

2. Refine technical descriptions while maintaining precise scientific meaning;

3. Improve the flow and readability of the writing;

4. Assist with formatting and stylistic consistency.

We explicitly confirm that the model was not involved in any aspects of research ideation, technical innovation, mathematical formulation, algorithm design, experimental implementation, or result analysis. All scientific contributions, including the proposed V-differences-based deceptive models (VDMs), theoretical foundations, experimental design, and conclusions are solely the intellectual product of the human authors.

The human authors maintained full oversight and responsibility throughout the writing process. All content generated with model assistance was rigorously reviewed, validated, and edited to ensure technical accuracy and alignment with the authors' original intent.

We affirm that the authors take complete responsibility for all content of this paper, including any portions that underwent linguistic refinement with model assistance. The model was not used to generate any core scientific content, and all key insights and contributions originate from the human authors.

**LLM Used:** DeepSeek-V3
**Scope of Use:** Writing assistance and language polishing only

