# OpenReview forum: "Strategic Deception in Deterministic Markov Decision Processes via Value Differences"
_ICLR.cc/2026/Conference — Submitted to ICLR 2026_

### Official Review · Reviewer_hQMd · 2025-11-02

**Soundness:** 3
**Presentation:** 3
**Contribution:** 2
**Rating:** 2
**Confidence:** 4

**Summary:**

The paper proposes a two-stage approach for performing deceptive planning over finite, deterministic Markov decision processes (MDPs), where the planning agent's objective is to navigate to its true goal while deceiving an external observer that is trying to infer the agent's true goal from amongst a set of candidate goals.In the first stage of the approach, the agent behaves deceptively up until a to-be-determined "last deceptive state" (LDS); in the second stage, the agent follows the minimum-cost path from the LDS to the true goal. The deception metrics used to perform deception combine the notions of exaggeration and ambiguity from previous works on DPP ([Masters & Sardina, 2017], [Savas et al., 2022]) with the maximum entropy-type observer model developed in [Ziebart et al., "Planning-based prediction for pedestrians", 2009], [Ziebart et al., "Modeling interaction via the principle of maximum causal entropy", 2010] and previously used for DPP in [Savas et al., 2022], [Chen et al., "Deceptive planning for resource allocation", 2024], and [Suttle et al., "Value of information-based deceptive path planning under adversarial interventions", 2025]. The notion of LDS is very closely related to the notion of "last deceptive point" developed in [Masters & Sardina, 2017]. The term "Value-differences-based deceptive model", or VDM, refers to the overall two-stage approach combining the maximum entropy observer model (referred to as the "V-differences" approach in this work) with previous deception metrics. A key motivation for the two-stage nature of the VDM approach is to avoid the inefficient deceptive behavior observed in the later stages of DPP trajectories generated using the Ambiguity Model (AM) for deception of [Liu et al., 2021], when deception is no longer useful and minimum cost planning is most efficient. Experimental results on gridworld problems are provided that validate the performance of the proposed approach against AM and indicate robustness to the deception hyperparameter of the VDM method.

**Strengths:**

The idea of separating the DPP process into two phases -- a deceptive phase and a min-cost/shortest-path phase -- is natural and intuitively appealing. Developing a formalism that provides a principled way to divide and structure the two phases is thus well-motivated. The proposed use of LDS as the dividing point between the two phases and the specific VDM variants proposed provide reasonable instantiations of such a formalism. Furthermore, this paper bridges two distinct lines of work on deceptive planning -- (i) the control-/planning-based approaches of [Ornik & Topcu, 2018], [Savas et al., 2022], and their successors; (ii) the DPP work of [Masters & Sardina, 2017] and its successors -- that, to my knowledge, no previous work has made a serious attempt to combine. Bridging these two lines of research is a worthy objective and is definitely of interest to their respective deception communities.

**Weaknesses:**

The paper suffers from the following weaknesses:
1. The LDS definition is highly similar to the "last deceptive point" (LDP) notion of [Masters & Sardina, 2017]. The primary difference appears to be the use of the observer model of [Savas et al., 2022], [Ziebart et al., 2009] to enable practical belief computation. This undermines claim (2) from the introduction.
2. Since the MDPs are finite and dynamics are deterministic, the underlying problem can be viewed as a standard DPP problem, as considered in [Savas et al., 2022] and [Masters & Sardina, 2017]. The setting considered in this work is on the one hand a special case of the general setting considered in [Savas et al., 2022], and on the other it is unclear how the LDS notion defined in this work is truly distinct from the LDP notion of [Masters & Sardina, 2017]. This undermines claim (3) from the introduction.
3. The proofs of the theoretical results follow immediately from the definitions, and the main theoretical contribution is thus the set of definitions proposed in Section 4. The theoretical contribution is thus minor, as it consists primarily of defining LDS and related objects, from which the main results follow easily.
4. The solution methods of [Savas et al., 2022], which apply to finite MDPs with stochastic dynamics, apply to the deterministic MDP setting considered in this work as a special case. Furthermore, the trajectories generated by these solution methods do not exhibit the wasteful late-trajectory behavior seen in the AM method of [Liu et al., 2021]. The methods of [Savas et al., 2022] are thus important baselines to compare with, yet the current work only compares with AM. This makes it difficult to assess the effectiveness of the proposed approach against these important and highly relevant baselines, weakening the experimental contribution of the paper.

**Questions:**

1. How does the proposed approach move "beyond the geometric constraints of traditional path planning" as claimed around lines 22-23?
2. In what way is the deterministic MDP setting addressed in your work more general than the problem setting considered in [Masters & Sardina, 2017]?
3. Is the deterministic MDP setting you consider a special case of the stochastic MDP setting considered in [Savas et al., 2022]? If not, how?
4. Definition 1 appears to assume an infinite-horizon discounted MDP, while the value function definition of line 158 has finite horizon $T$. Which setting do you consider?
5. In what way does the observer model described in Section 3, which holds for stochastic problems in general (see [Ziebart et al., 2009-2010], [Savas et al., 2022]), exhibit "natural alignment with deterministic MDPs", as claimed on lines 167-168?
6. The motivation of LDS in Section 4 appears to rest on the *assumption* that a two-phase approach is best or at least preferable to other possible approaches; is this accurate, or is there an *a priori* reason why the two-phase approach is best/preferable?
7. Can you elaborate on the exact differences between the LDS of Definition 4 and the LDP of Definition 6 in [Masters & Sardina, 2017]?
8. How are Definitions 6 and 7 different?
9. On lines 314-315 it is claimed that you "establish that the optimal decoy goal state is $g^*$" -- where is this established?
10. Around line 317 it is stated that "$g^* $ possesses the strongest deceptive potential among all fake goal states at $d^*_c$" -- where is this shown?
11. Aside from the use of the maximum entropy observer model developed in [Ziebart et al., 2009-2010] and used for DPP in [Savas et al., 2022], what are the primary differences between the deception notions considered in Section 5 and those considered in [Masters & Sardina, 2017] and [Liu et al., 2021]? How do the notions of exaggeration and ambiguity that you use relate to the corresponding notions in [Savas et al., 2022]?
12. Given that the observer model of Section 3 is a core component of your approach and is based on [Savas et al., 2022], it would be useful to compare your approach with the methods proposed in [Savas et al., 2022] -- are there any issues preventing such a comparison?

---

> ### Author Response · Authors · 2025-11-19
>
> We sincerely thank you for your detailed and constructive feedback. Your questions have helped us clarify several key aspects of our work. Below, we address each of your points.
>
> **Q1: How does the proposed approach move "beyond the geometric constraints of traditional path planning" as claimed around lines 22-23?**
> **Q2: In what way is the deterministic MDP setting addressed in your work more general than the problem setting considered in [Masters & Sardina, 2017]?**
>
> First, while our Last Deceptive State (LDS) is conceptually similar to the Last Deceptive Point (LDP) from [Masters & Sardina, 2017] in that both seek a "turning point," their foundations are fundamentally different. LDP is defined within the geometric context of path planning and is computed based on cost-differences using algorithms like A*. In contrast, LDS is defined in the value function space of an MDP. Its determination relies on the MDP's value function, not on geometric paths. This distinction allows our framework to extend to problem domains where LDP is not applicable:
>
> 1.  **Abstract Nature of States:** In our framework, a "state" is not limited to physical coordinates (e.g., (x, y)). It can be any discrete node representing the world's condition. For instance, in a cybersecurity "honeypot" application, a state could represent an attacker's exploration depth and access level within a decoy system, while an "action" could be the system revealing new fabricated information. Similarly, in deceptive business decisions, there is no concept of geometric cost-differences.
>
> 2.  **Flexibility of Rewards:** Traditional path planning typically involves reaching a fixed goal, but the MDP framework allows for a much richer reward structure. In a DMDP, different states can have varying values based on their importance, which directly impacts the V-differences calculation. This is a fundamental departure from the DPP approach in [Masters & Sardina, 2017].
>
> This was an interesting research direction we discovered while addressing the limitations of the AM: exploring the connection between control/planning-based methods (e.g., [Ornik & Topcu, 2018], [Savas et al., 2022]) and classic deceptive path planning ([Masters & Sardina, 2017]). We questioned whether the LDP concept could be generalized to DMDPs and, potentially, to broader classes of MDPs. We believe bridging these two research lines is a meaningful and intriguing objective.
>
> **Q3: Is the deterministic MDP setting you consider a special case of the stochastic MDP setting considered in [Savas et al., 2022]? If not, how?**
>
> Yes, mathematically, a deterministic MDP is a special case of a stochastic MDP where state transitions are not random. However, the core of our work differs from that of DDM. DDM focuses on generating a globally optimal policy that maximizes deception in a stochastic environment, resulting in a complete policy covering the entire state space. Our work, however, uncovered that when transitions are deterministic, DMDPs exhibit unique and interesting characteristics regarding deception. This led us to develop a two-phase planning method that is intuitive, effective, and computationally efficient.
>
> **Q4: Definition 1 appears to assume an infinite-horizon discounted MDP, while the value function definition of line 158 has finite horizon T. Which setting do you consider?**
>
> Thank you for helping us clarify this crucial point in our model's definition. We use an infinite-horizon discounted MDP framework with absorbing goal states. When the agent reaches any goal state, it transitions to itself with 100% probability on all future time steps and receives zero reward.
>
> As you suggested, we have revised the paper accordingly. We have changed $T$ to $\infty$ on line 158 and added the sentence, "We assume that all goal states are absorbing, i.e., $P(g, a, g) = 1$ for all $g \in G$" on line 209.
>
> **Q5: In what way does the observer model described in Section 3, which holds for stochastic problems in general (see [Ziebart et al., 2009-2010], [Savas et al., 2022]), exhibit "natural alignment with deterministic MDPs", as claimed on lines 167-168?**
>
> This is because [Savas et al., 2022] cites the conclusion from [Ziebart et al., 2009-2010] that V-differences can be used as an approximation "when the transition randomness has a limited effect on the agent’s behavior and the discount factor $\gamma$ is large enough." This remains an approximation in stochastic MDPs: the exact threshold for valid approximation may vary. In our DMDP setting, state transitions are deterministic by definition, which perfectly fits this condition. To make this clearer for the reader, we have updated the relevant statement.
>
> On line 187 of the paper, we have changed "The probability calculation of goal states based on V-differences exhibits a natural alignment with DMDPs" to "The probability calculation based on V-differences is well-suited for DMDPs given their deterministic transitions."

---

> ### Author Response · Authors · 2025-11-19
>
> **Q6: The motivation of LDS in Section 4 appears to rest on the assumption that a two-phase approach is best or at least preferable to other possible approaches; is this accurate, or is there an a priori reason why the two-phase approach is best/preferable?**
>
> Structuring the deception process as a two-phase model is a deliberate design choice aimed at solving the problem observed in the AM, where the agent continues to "deceive" inefficiently even when it is close to the true goal. This led us to consider whether the entire deceptive plan could be segmented, allowing the agent to focus on deception in the first phase and on optimality in the second. This approach is guided by the principle that "an agent, after its true intention has been and will continue to be exposed, should complete its task as efficiently as possible."
>
> This line of reasoning prompted us to investigate "how to find such a turning point." We were excited to discover that DMDPs solved with value iteration exhibit properties very similar to those in DPP. This encouraged us to explore these properties further and to imagine whether such properties might also exist for more general, stochastic MDPs.
>
> Therefore, the two-phase approach was not an arbitrary choice but a targeted solution proposed in response to the identified shortcomings of the AM.
>
> **Q7: Can you elaborate on the exact differences between the LDS of Definition 4 and the LDP of Definition 6 in [Masters & Sardina, 2017]?**
>
> 1.  **Different Problem Scopes:** The core of LDP is path cost, whereas the core of LDS is value (long-term return). Treating a DMDP merely as a path-planning problem can lead to a loss of critical information. For example, consider an environment with an intermediate, non-absorbing state that provides a positive reward. The LDP framework, being based entirely on path cost, has no mechanism to understand or evaluate the value of this intermediate state. In contrast, the LDS, by incorporating a reward function and value iteration, can handle this situation effectively. The value iteration process propagates the positive reward from the intermediate state throughout the environment, reflecting it in the value function V(s). Thus, LDS can recognize a seemingly circuitous but necessary path as rational or even optimal. LDS addresses a broader and richer decision-making problem than LDP.
>
> 2.  **Different Computational Methods:** The computation of LDP is a two-stage, pairwise, and local process. As you know, the method in [Masters & Sardina, 2017] first identifies a "single fake goal" $g_{min}$ from all fake goals through path cost comparison. Then, it determines a geometric balance point between the real goal $g_r$ and this $g_{min}$ via algebraic operations on their optimal path costs. In contrast, the computation of LDS does not pre-select any goal. Instead, it computes a value function $V(s, g_i)$ over the entire state space for all goals $g_i \in G$. Each $V(s, g_i)$ forms a "value field" in the space. The position of the optimal LDS ($d_c^*$) is determined by the topological boundary formed by the superposition and competition of all these value fields.
>
> Your point in the "Weaknesses" section—that "Since the MDPs are finite and dynamics are deterministic, the underlying problem can be viewed as a standard DPP problem"—has prompted deep reflection. While this may hold at a graph-theoretic level (the state transition graph of a DMDP can be seen as a graph), we argue that this abstraction loses the crucial "value" dimension provided by the MDP framework. This is precisely the fundamental difference between LDS and LDP. LDP seeks a "geometrically" optimal solution (shortest path), whereas LDS seeks a "decision-theoretically" optimal one (highest long-term value). Compared to the LDP method, which is suited for classic deceptive path planning, LDS introduces the concepts of reward functions and value iteration. It can be seen as a transition from "classic path planning" to the "reinforcement learning domain," and from "true/fake goal recognition" to "true/fake reward function recognition." We believe exploring the generalizability of the LDP concept to DMDPs and beyond is a meaningful and interesting goal.
>
> **Q8: How are Definitions 6 and 7 different?**
>
> Thank you for this question, which has helped us identify an error in our wording that may have hindered your understanding, for which we apologize. On line 270 of the paper, we have changed "any action" to "there exists an action."
>
> A state is on the "edge" of the deceptive set if *there exists* an action that improves the agent's value with respect to the real goal state and *immediately forces it to exit* the deceptive set. In essence, any state in $D_e$ can serve as the LDS for some trajectory. Definition 7, which defines the critical deceptive states set $D_c$, describes the *optimal* LDS candidates, for which *any* action that improves the agent's position toward $g_r$ will make the next state truthful.

---

> ### Author Response · Authors · 2025-11-19
>
> **Q9: On lines 314-315 it is claimed that you "establish that the optimal decoy goal state is $g^\*$" -- where is this established?**
> **Q10: Around line 317 it is stated that "$g^\*$ possesses the strongest deceptive potential among all fake goal states at $d_c^\*$" -- where is this shown?**
>
> Thank you for pointing this out. Our use of the word ‘establish’ on line 315 was likely imprecise, and the phrase ‘deceptive potential’ on line 317 was not rigorous enough.
>
> The intent of this paragraph was to convey that we **identify** the optimal decoy goal, $g^\*$. This $g^\*$ is the fake goal that has the highest posterior probability in the observer's eyes at the moment the agent reaches the optimal LDS, $d_c^\*$. A good "Exaggeration" strategy should aim to make the observer most confident that the agent is heading toward a specific fake goal during the early and middle stages of the plan.
>
> To make our statements clearer, we have made the following revisions to the paper:
> 1.  On line 315, we have replaced ‘establish’ with ‘identify’.
> 2.  On line 317, we have replaced the sentence “$g^\*$ possesses the strongest deceptive potential among all fake goal states at $d_c^\*$.” with “$g^\*$ represents the decoy goal state with the maximum posterior probability given the agent's state $d_c^\*$, making it the decoy goal state most likely to be mistaken for the true one.”
>
> **Q11: Aside from the use of the maximum entropy observer model developed in [Ziebart et al., 2009-2010] and used for DPP in [Savas et al., 2022], what are the primary differences between the deception notions considered in Section 5 and those considered in [Masters & Sardina, 2017] and [Liu et al., 2021]? How do the notions of exaggeration and ambiguity that you use relate to the corresponding notions in [Savas et al., 2022]?**
>
> Section 5 introduces three methods based on two concepts:
>
> One method is based on the concept of **Ambiguity**:
> 1.  **Ambiguity-VDM:** This method is divided into two stages. In the first stage (from the start state to $d_c^\*$), the action selection policy is identical to that of the AM model. In the second stage (from $d_c^\*$ to $g_r$), the agent follows the optimal policy to the true goal. This method was designed to analyze whether the two-stage approach significantly helps in avoiding the persistent deception problem of AM.
>
> Two methods are based on the concept of **Exaggeration**:
> 1.  **Parameterized-Exaggeration-VDM:** In the deceptive phase of this method, a parameter $\sigma$ controls the agent's tendency to move away from the true goal $g_r$ and toward the fake goal $g^\*$ while progressing toward $d_c^\*$.
> 2.  **Decoy-First-Exaggeration-VDM:** In the deceptive phase of this method, the agent first travels to $g^\*$, then to $d_c^\*$, before entering the truthful phase to reach $g_r$. This is inspired by the exaggeration concept in [Masters & Sardina, 2017] (specifically, $\pi_1$), where the agent first visits a fake goal.
>
> Furthermore, the notion of deception in [Savas et al., 2022] involves incorporating the posterior probability of goal states into a linear program for solving. In contrast, the deception concepts in our Section 5 are based entirely on V-value calculations. For instance, the policy $a^\* = \arg\max_{a \in A^+(s)} \left[ \Delta V_{d_c^\*}(s, s') + \sigma \cdot \left( \Delta V_{g^\*}(s, s') - \Delta V_{g_r}(s, s') \right) \right]$ indicates that during the first stage, the agent has a tendency of degree $\sigma$ to move closer to the fake goal $g^\*$.
>
> **Q12: Given that the observer model of Section 3 is a core component of your approach and is based on [Savas et al., 2022], it would be useful to compare your approach with the methods proposed in [Savas et al., 2022] -- are there any issues preventing such a comparison?**
>
> Thank you for this suggestion. Initially, we did not compare VDM with DDM because we considered that DDM uses a maximum entropy value iteration method, while we use a more traditional value iteration (setting $\alpha \to 0$), resulting in different observer models. Your reminder prompted us to recognize the necessity of this comparison.
>
> We have since conducted supplementary experiments comparing VDM and DDM. We aligned the observer models for both and used parameter settings for DDM consistent with those in the DDM vs. DPP comparison in [Savas et al., 2022]. We found that each method has its own advantages across different metrics. You can find the results of this new comparison in the updated version of our paper.
>
> Thank you again for your high-quality questions; they have been of great benefit to us.
>
> We hope these responses have addressed your concerns. We’re happy to provide further clarification if needed. Thank you for your careful review—if our response has resolved your concerns, we would sincerely appreciate your consideration in updating the score.

---

> > ### Author Response · Authors · 2025-11-28
> >
> > Dear Reviewer hQMd,
> >
> > We sincerely hope this message finds you well. We would like to express our deepest gratitude for the time and effort you have dedicated to reviewing our submission and for providing such thoughtful and constructive feedback.
> >
> > We have carefully addressed all the questions and concerns raised in your reviews, and our detailed responses have been submitted through the rebuttal system. As the author-reviewer discussion period is approaching its conclusion (with less than one week remaining before the deadline), we wanted to kindly reach out to ensure that you have had the opportunity to review our responses.
> >
> > We understand that you have many responsibilities and reviewing demands on your time. However, if you have any remaining questions, concerns, or would like us to clarify any aspect of our rebuttal, we would be more than happy to provide additional information or elaboration. We are committed to engaging in a productive dialogue to address any outstanding issues.
> >
> > Your feedback has been invaluable in helping us improve our work, and we would greatly appreciate the opportunity to know whether our responses have adequately addressed your concerns. If our responses have resolved the issues you raised, we would be deeply grateful if you could kindly consider acknowledging this in the discussion thread or updating your assessment accordingly.
> >
> > Once again, thank you very much for your careful consideration of our work. We look forward to hearing from you at your earliest convenience.
> >
> > With sincere appreciation and respect,
> >
> > The Authors

---

### Official Review · Reviewer_qxpt · 2025-11-03

**Soundness:** 3
**Presentation:** 3
**Contribution:** 3
**Rating:** 6
**Confidence:** 4

**Summary:**

The paper presents a framework for deceptive planning in Markov decision processes (MDPs) with deterministic transition functions. Specifically, instead of defining the deception over reward and Q functions, the authors define the likelihood of a goal state based on differences of the value functions (V-differences). Then, they develop a concept called the last deceptive state (LDS), a method to compute the optimal LDS, and deceptive models. The approach can compute trajectories with a lower cost compared to an existing method called the ambiguity model, while having a much lower likelihood of reaching the targeted goal state through the trajectory.

Overall, the paper develops a new deceptive planning concept based on LDS instead of having it as entropy-based, which improves efficiency in terms of the final cost, as it does not rely on acting randomly to increase entropy.

I believe the paper develops an interesting framework for deceptive planning, and the approach is theoretically sound and improves over an existing deceptive model, AM, in practice. I believe the paper can be accepted with minor revisions.

**Strengths:**

Overall, the paper develops a new deceptive planning concept based on LDS instead of having it as entropy-based, which improves efficiency in terms of the final cost, as it does not rely on acting randomly to increase entropy. The metholodgy is sound and the practical examples show that the resulting trajectories look more reasonable while still being deceptive compared to the honest trajectories.

**Weaknesses:**

The framework requires computing the value functions for each state and action in the MDP. It also assumes perfectly rational observers, which may be restrictive in terms of applying the framework.

**Questions:**

What is the key intuition behind introducing LDS compared to existing concepts like the Last Deceptive Point (LDP), which the authors mentioned in the related work section? How does LDS improve upon or generalize the LDP conceptually and practically?

Could the authors explicitly describe how the LDS formulation directly addresses the inefficiency of AM? I believe it's because of not needing to act randomly to increase entropy, but the authors can further clarify it.

The authors assume that their formulation assumes a perfectly rational observer, i.e., \alpha=0, which makes the optimal policy deterministic, and the value function does not need to have a softmax operator over actions. How would the framework change if \alpha > 0, and is LDS guaranteed to exist with \alpha > 0?

The authors proved that an LDS always exists. Is the LDS unique, or can it change based on the agent's policy and trajectory?

What is the computational complexity of LDS in terms of the number of goal states and the states in the MDP? Finally, is it possible to approximately compute LDS in large state spaces, or does the current algorithm only support computing LDS with precomputed value functions?

Are there theoretical guarantees on deception strength or cost ratios for either variant, i.e., can we bound the increase in trajectory cost relative to the honest policy?

---

> ### Author Response · Authors · 2025-11-19
>
> We sincerely thank you for your positive assessment and insightful questions. We are glad you found our framework interesting and theoretically sound. Your feedback is invaluable for refining our work, and we would like to address your questions and the weaknesses you identified.
>
> **Q1: What is the key intuition behind introducing LDS compared to existing concepts like the Last Deceptive Point (LDP), which the authors mentioned in the related work section? How does LDS improve upon or generalize the LDP conceptually and practically?**
>
> The key intuition for introducing the Last Deceptive State (LDS) instead of directly adopting the Last Deceptive Point (LDP) is to **generalize deceptive planning from a framework applicable only to path planning to a more universal, value-based sequential decision-making framework.**
>
> Both LDS and the LDP from [Masters & Sardina, 2017] seek to identify a "turning point." However, their foundations are fundamentally different:
>
> *   **Geometric vs. Value-Based:** LDP is defined within the geometric context of path planning, calculated using cost-differences from an A\* search. In contrast, LDS is defined in the value function space of an MDP. Its determination relies on the MDP's value functions, not on geometric paths. This allows LDS to extend to problem domains where LDP is not applicable.
>
> *   **Abstractness of States:** In our framework, a "state" is not limited to physical coordinates (e.g., (x, y)). It can be any discrete, abstract node. For instance, in a cybersecurity "honeypot" application, a "state" could represent an attacker's exploration depth and access level, while an "action" is the system revealing new fabricated information. Similarly, in deceptive business decisions, there is no concept of geometric cost-differences.
>
> *   **Flexibility of Rewards:** Traditional path planning often assumes all goals are equivalent. The MDP framework, however, allows for a much richer reward structure. Different goals can have different values based on their importance, which directly impacts the V-differences calculation. This is another aspect where LDS is more broadly applicable than LDP, which, as in DPP [Masters & Sardina, 2017], treats all goals equally.
>
> In summary, **LDS elevates the core metric from "cost-differences" to "V-differences."** The value function V(s) captures the long-term expected return from a state, incorporating costs, rewards, and the discount factor γ. This generalization expands the applicability of the two-phase deception concept from path planning to a much broader class of sequential decision-making problems.
>
> **Q2: Could the authors explicitly describe how the LDS formulation directly addresses the inefficiency of AM? I believe it's because of not needing to act randomly to increase entropy, but the authors can further clarify it.**
>
> Your understanding is perfectly correct. The fact that our framework **"does not rely on acting randomly to increase entropy"** is precisely how the LDS formulation addresses the inefficiency of the Ambiguity Model (AM).
>
> The inefficiency of AM stems from its core strategy of "maximizing entropy at every decision step." This leads to ineffective deception in the later stages of a task. As the agent approaches its real goal ($g_r$), the observer's posterior probability for $g_r$ will inevitably surpass that of all fake goals, regardless of the agent's actions. At this point, the observer has effectively seen through the deception. Despite this, AM remains driven by its entropy-maximization objective, forcing the agent to take redundant and costly actions even after its intent has been exposed.
>
> Our LDS framework fundamentally solves this by introducing a **"two-phase planning" paradigm.** The core contribution of LDS is to identify and compute the effective boundary of deception—the optimal Last Deceptive State ($d_c^\*$).
> *   **Before $d_c^\*$ (Deceptive Phase):** Deception is viable and meaningful. The agent executes a deceptive strategy (ambiguity or exaggeration).
> *   **After $d_c^\*$ (Truthful Phase):** The agent's intent is considered exposed. It switches to the optimal policy to reach $g_r$ as quickly as possible, minimizing its exposure time.
>
> This strategic separation not only avoids the significant cost wastage seen in AM's late-stage behavior but also makes the entire deceptive trajectory more intelligent and efficient.

---

> ### Author Response · Authors · 2025-11-19
>
> **Weakness & Q3: The framework requires computing the value functions for each state and action in the MDP. It also assumes perfectly rational observers... How would the framework change if α > 0, and is LDS guaranteed to exist with α > 0?**
>
> We appreciate you raising this important point. While our paper assumes a perfectly rational observer (α → 0) to clearly establish the theoretical foundations in a deterministic setting (aligning with classic rational planning models like Ramírez & Geffner, 2010), the framework is adaptable to non-perfectly rational observers.
>
> If we relax this assumption to **α > 0**, the observer becomes "Boltzmann-rational"—they assume the agent prefers higher-value actions but does not always choose the optimal one. To deceive such an observer, our framework would require the following adaptation:
>
> *   **Value Iteration with Soft Bellman Equation:** To model the observer's reasoning, the agent's offline computation of value functions $V_g(s)$ for all goals $g$ must switch from the standard Bellman optimality equation to the **soft Bellman optimality equation**:
>     $V_g(s) = \alpha \cdot \log\left( \sum_{a \in A} \exp(Q_g(s, a) / \alpha) \right)$
>     The value function $V_g(s)$ now represents a "smoothed" maximum over all possible actions, and the optimal policy $\pi_g^\*(s)$ becomes stochastic (a Boltzmann distribution over Q-values).
>
> Does LDS still exist when α > 0? Yes, the existence of an LDS is guaranteed because the core logic remains intact:
> 1.  **Definition of LDS is Unchanged:** The definition of a deceptive state—$Pr(g_r | s) ≤ Pr(g | s)$ for some fake goal $g$—still holds. This condition is equivalent to $ΔV_{g_r}(s) ≤ ΔV_g(s)$, which is well-defined using the new soft value functions.
> 2.  **Initial State is Still Deceptive:** At the initial state $s_1$, the value gain $ΔV_g(s_1) = V_g(s_1) - V_g(s_1) = 0$ for all goals. Thus, $ΔV_{g_r}(s_1) = ΔV_g(s_1)$, satisfying the condition for a deceptive state regardless of the value of α.
> 3.  **Existence Proof:** Since any trajectory begins in a deceptive state ($s_1$) and ends in a truthful one ($g_r$), there must exist a "last" deceptive state along this path, after which all subsequent states are truthful.
>
> Therefore, even for α > 0, the LDS concept remains robust. The computation of the optimal LDS, $d_c^\* = \arg\max_{s \in D} V_{g_r}(s)$, also remains applicable, though the set of deceptive states $D$ and the value function $V_{g_r}(s)$ would be based on the new soft value calculations.
>
> We do note, as discussed in [Savas et al., 2022], that as α becomes very large, the agent's policy approaches pure randomness, at which point the concept of "deception" may lose its meaning.
>
> **Q4: The authors proved that an LDS always exists. Is the LDS unique, or can it change based on the agent's policy and trajectory?**
>
> This is an excellent question that touches on a key nuance of our framework. The LDS is **not globally unique** as it depends on the specific trajectory; however, our proposed **optimal LDS ($d_c^\*$) is a global concept** that serves as the core target for our two-phase planning.
>
> 1.  **LDS for a Given Trajectory (LDS_ζ): Trajectory-Dependent.** As per Definition 4 in our paper, the Last Deceptive State is defined with respect to a specific trajectory ζ. It is the last state in that sequence that is deceptive. Since different policies produce different trajectories (i.e., different state sequences), the LDS for each of these trajectories can naturally be different.
>
> 2.  **Optimal LDS ($d_c^\*$): Globally Determined (but not necessarily a single state).** One core contribution of our work is to answer, "Which LDS is the best one to aim for?" We address this by defining the optimal Last Deceptive State, $d_c^\* = \arg\max_{s \in D} V_{g_r}(s)$. This is a global optimization that depends on the problem definition (state space S, goal set G, initial state $s_1$, and value functions V), not on any specific trajectory. It represents the "most advantageous" deceptive endpoint for the agent before it must switch to its true plan. In highly symmetric environments, it is theoretically possible for multiple states to share the same maximal $V_{g_r}(s)$ value. This is precisely why we defined the Critical Deceptive States Set, $D_c$, which is the set of all such optimal states. From a planning perspective, any of these states can be chosen as the target, as they are equivalently optimal.
>
> We hope this clarifies the relationship between the trajectory-dependent nature of a generic LDS and the global optimality of our proposed $d_c^\*$.

---

> ### Author Response · Authors · 2025-11-19
>
> **Q5: What is the computational complexity of LDS in terms of the number of goal states and the states in the MDP? Finally, is it possible to approximately compute LDS in large state spaces...?**
>
> Thank you for these critical questions on complexity and scalability.
> 1.  **Computational Complexity:** The computation of the optimal LDS ($d_c^\*$) has two main stages:
>     *   **Offline Value Iteration:** We first compute the optimal value function $V_g(s)$ for each of the $|G|$ goals across the entire state space $|S|$. The complexity for a single goal is O(k * |S| * |A|), where k is the number of iterations to convergence. Thus, this stage's total complexity is **O(|G| * k * |S| * |A|)**.
>     *   **Identifying $d_c^\*$:** After computing the value functions, we identify the deceptive state set $D$ by iterating through all states (O(|S| * |G|)) and then find the state in $D$ that maximizes $V_{g_r}(s)$ (O(|D|)).
>     The overall complexity is dominated by the first stage (value iteration).
>
> 2.  **Approximation in Large State Spaces:** For extremely large or continuous state spaces where tabular methods are infeasible, the LDS concept can be extended using function approximation (e.g., deep neural networks).
>     *   **Approximate Value Functions:** One could train a separate neural network, $V_g(s; \theta_g)$, for each goal to approximate the value function, using algorithms like DQN.
>     *   **Approximate $d_c^\*$:** Finding $d_c^\*$ then becomes a constrained optimization problem: $d_c^\* \approx \arg\max_s V_{g_r}(s; \theta_{g_r})$ subject to $s$ being a deceptive state. This could potentially be solved with gradient-based methods, for example, by performing gradient ascent on $V_{g_r}(s; \theta_{g_r})$ from $s_1$ while using a projection or penalty to remain within the approximate boundary of the deceptive set $D$.
>
> Our current algorithm is indeed model-based and relies on pre-computed value functions to establish the theoretical foundation. However, we believe the core idea of identifying a deceptive endpoint for two-phase planning has strong potential for extension to larger, more complex problems. We are grateful for the opportunity to discuss these exciting future directions.
>
> **Q6: Are there theoretical guarantees on deception strength or cost ratios for either variant...?**
>
> This is a challenging and important question. To be transparent, our current work does not provide formal mathematical bounds on deception strength or trajectory cost ratios. Our focus has been on establishing the theoretical foundation of the LDS framework (existence, optimality) and demonstrating its empirical effectiveness.
>
> However, we can explain the properties of the variants:
> *   **Trajectory Cost:** It is difficult to provide a simple cost ratio relative to "Cost(Honest)" because the path in Stage 1 is highly dependent on the environment's geometry (obstacles, goal locations). The paths for E-VDM(σ) and A-VDM are generated heuristically, making their costs difficult to analyze theoretically. However, our framework strategically controls the total cost by ensuring Stage 2 (from $d_c^\*$ to $g_r$) is optimally efficient.
> *   **Deception Strength:** A uniform theoretical guarantee on deception strength (i.e., how low $Pr(g_r|s)$ can be) is also challenging, as it depends on the relative differences between the various goal value functions, which are instance-specific. It is also influenced by the observer model.
>
> Overall, while we cannot currently offer universal cost-ratio bounds, our framework's design inherently controls costs by ensuring the second phase is optimal. For deception, our VDM variants are designed at an algorithmic level to directly optimize deception metrics, and experiments show this is effective and stable. We believe that if hard constraints on cost or deception are required, one could formulate this as a problem to be solved with methods like linear programming.
>
> We hope this response has resolved your concerns. Thank you again for your careful review. We’re happy to provide further clarification if needed. Thank you for your careful review—if our response has resolved your concerns, we would sincerely appreciate your consideration in updating the score.

---

> > ### Author Response · Authors · 2025-11-28
> >
> > Dear Reviewer qxpt,
> >
> > We sincerely hope this message finds you well. We would like to express our deepest gratitude for the time and effort you have dedicated to reviewing our submission and for providing such thoughtful and constructive feedback.
> >
> > We have carefully addressed all the questions and concerns raised in your reviews, and our detailed responses have been submitted through the rebuttal system. As the author-reviewer discussion period is approaching its conclusion (with less than one week remaining before the deadline), we wanted to kindly reach out to ensure that you have had the opportunity to review our responses.
> >
> > We understand that you have many responsibilities and reviewing demands on your time. However, if you have any remaining questions, concerns, or would like us to clarify any aspect of our rebuttal, we would be more than happy to provide additional information or elaboration. We are committed to engaging in a productive dialogue to address any outstanding issues.
> >
> > Your feedback has been invaluable in helping us improve our work, and we would greatly appreciate the opportunity to know whether our responses have adequately addressed your concerns. If our responses have resolved the issues you raised, we would be deeply grateful if you could kindly consider acknowledging this in the discussion thread or updating your assessment accordingly.
> >
> > Once again, thank you very much for your careful consideration of our work. We look forward to hearing from you at your earliest convenience.
> >
> > With sincere appreciation and respect,
> >
> > The Authors

---

### Official Review · Reviewer_Soss · 2025-11-05

**Soundness:** 2
**Presentation:** 3
**Contribution:** 2
**Rating:** 4
**Confidence:** 3

**Summary:**

This paper introduces the V-differences-based Deceptive Models (VDMs), a novel framework for creating autonomous agents capable of strategic deception in Deterministic Markov Decision Processes (DMDPs). They addresses a critical limitation of previous approaches, such as the Ambiguity Model (AM), to provide a principled balance between the goals of deception and efficiency.

**Strengths:**

- The authors identify that AM's strategy of blindly maximizing entropy at every step leads to a large number of redundant and ineffective deceptive actions in the later stages of a task, resulting in significant efficiency loss once the agent's intention is effectively revealed.
- They introduce the Last Deceptive State (LDS), which serves as an intermediate transitional state that clearly separates the agent’s strategy into two phases—deceptive and truthful. The work is theoretically grounded and explained
- They show results in path planning environments showing that VDMs outperform AM.

**Weaknesses:**

-  Although the work is effective in deterministic settings, the proposed models are not easily extendable to stochastic or partially observable environments.
- I am not sure how applicable these findings are to the real-world where there is a lot of uncertainty and more complex environments
- The formalism assumes a a fully rational observer which also limits real word application, as humans often behave sub-optimally

**Questions:**

- Why did you choose value differences (V-differences) instead of Q-differences as the foundation for deception modeling?
- How does your definition of deception align with human's intuitive notion of deception?
- Are there real-world settings where your system would be applicable?

---

> ### Author Response · Authors · 2025-11-19
>
> We sincerely thank you for your insightful feedback and for recognizing the core contributions of our work. Your questions have helped us to further clarify our motivations and the potential impact of our framework. We would like to address the weaknesses and questions you raised.
>
> **Weakness 1: Although the work is effective in deterministic settings, the proposed models are not easily extendable to stochastic or partially observable environments.**
>
> We agree that extending our models to stochastic and partially observable environments is an important direction for future work. Our decision to first focus on Deterministic Markov Decision Processes (DMDPs) was a strategic choice for two primary reasons:
>
> 1.  **Isolating and Solving the Core Problem:** Our primary motivation was to address a fundamental flaw in prior methods like the Ambiguity Model (AM)—namely, its continued and inefficient pursuit of entropy maximization even after its intent has been exposed. The deterministic nature of DMDPs allowed us to precisely isolate and identify this strategic deficiency without the confounding variable of environmental randomness. Had we started in a stochastic setting, it would have been difficult to distinguish whether inefficiencies stemmed from the strategy itself or from environmental chance. Therefore, validating our two-phase planning framework in DMDPs was a necessary first step to prove its core value.
>
> 2.  **Building a Solid Theoretical Foundation:** One of our key theoretical contributions is the formalization of the Last Deceptive State (LDS), including its existence proof and the method for computing the optimal LDS ($d_c^\*$) (Theorem 3). The rigorous derivation of these theories relies on the deterministic properties of the value function in a DMDP. This provides a solid theoretical bedrock for our work. As you noted, extending this to stochastic settings is non-trivial, which underscores the importance of first establishing this foundation. Without it, generalization to more complex environments would lack theoretical grounding.
>
> We envision the path forward for extension as follows:
>
> *   **Extension to Stochastic MDPs:** The core idea of a two-phase plan (deceptive and truthful) likely still applies. The key adaptation would be in the definition of a "truthful state." In a stochastic environment, an action can lead to a distribution of subsequent states. Thus, the definition would need to shift from a deterministic next state to an *expected belief* over the distribution of next states. Similarly, computing the optimal LDS would evolve from an `argmax` over deterministic values to an optimization over expected values. We believe the theoretical framework established in this paper provides a firm basis for tackling these challenges.
>
> *   **Extension to POMDPs:** In partially observable settings, both the agent and the observer face uncertainty. Our LDS concept could be naturally generalized to a **"Last Deceptive Belief State."** Planning would occur in belief space, with the objective being to plan a trajectory to the boundary of a "truthful belief region" and then execute an optimal policy from that boundary.
>
> We are grateful for the opportunity to discuss these exciting future directions with you.

---

> ### Author Response · Authors · 2025-11-19
>
> **Weakness 2 & Q3: I am not sure how applicable these findings are to the real-world where there is a lot of uncertainty and more complex environments. Are there real-world settings where your system would be applicable?**
>
> We appreciate you raising this important point about real-world applicability. We acknowledge that many real-world scenarios involve stochasticity. However, the DMDP framework is a reasonable and powerful abstraction for numerous important real-world applications where the environment is highly structured or controlled. The VDM framework is applicable to the domains that can be modeled as a sequential decision-making problem with discrete states and actions. Here are a few concrete examples:
>
> 1.  **Strategic Planning:**
>     *   **Adversarial Games & Military Feints:** Action planning in zero-sum games or military contexts is often based on deterministic models of the environment. An AI agent planning a feint could use our VDM framework to execute a sequence of moves that appear to target Point A (a decoy) while its true objective is Point B. This strategy can effectively mislead an opponent to gain a strategic advantage.
>     *   **VIP Escort Missions:** An autonomous convoy could employ deceptive pathing to shake off potential trackers. The LDS would represent the last possible moment to switch from a meandering, deceptive route to a direct, high-speed path to the secure destination.
>
> 2.  **Cybersecurity and Honeypots:**
>
>     A honeypot system can be modeled as a DMDP where a "state" represents an attacker's exploration depth and access level within the decoy system, and an "action" is the system revealing new fabricated information (e.g., a fake password file or an open port). The system can use VDM to plan a deceptive "path," guiding the attacker to believe they are approaching a high-value decoy target (e.g., the "Financial Database"). Our LDS, in this context, represents the point of no return where the attacker is fully committed to the trap. Once the LDS is reached, the system can switch from the "luring" phase to a "full surveillance and forensics" phase.
>
> **Weakness 3: The formalism assumes a fully rational observer which also limits real-world application, as humans often behave sub-optimally.**
>
> This is an excellent point. While our paper assumes a perfectly rational observer ($\alpha \to 0$) for clarity and consistency with classic rational planning models, our framework's core theoretical contributions—the definition of LDS, its existence proof (Corollary 1), and the method for computing the optimal LDS (Theorem 3)—remain valid and applicable for sub-optimal observers.
>
> Here is why the framework is robust to this assumption:
>
> If we relax the assumption of perfect rationality (i.e., for $\alpha > 0$), the observer becomes "Boltzmann-rational." They believe the agent *tends* to choose higher-value actions but does not *always* choose the optimal one. To deceive such an observer, our framework's value iteration would be updated: the agent, when modeling the observer's reasoning, would switch from the standard Bellman optimality equation to the **soft Bellman equation**:
> $V_g(s) = \alpha \cdot \log\left( \sum_{a \in A} \exp(Q_g(s, a) / \alpha) \right)$.
> The value function $V_g(s)$ now represents a "soft" maximum over possible actions, and the optimal policy $\pi^\*_g(s)$ becomes stochastic (a Boltzmann distribution over Q-values).
>
> However, even with this change, the existence of the LDS is guaranteed because the core logic holds:
>
> 1.  **Definition of LDS is Unchanged:** The LDS is defined based on posterior probabilities. A state $s$ is deceptive if, for some fake goal $g$, $\text{Pr}(g_r | s) \leq \text{Pr}(g | s)$. Using the probability formula with $\alpha > 0$, this is equivalent to $\Delta V_{g_r}(s) \leq \Delta V_g(s)$.
>
> 2.  **The Initial State is Still Deceptive:** At the start state $s_1$, the value gain $\Delta V_g(s_1) = V_g(s_1) - V_g(s_1) = 0$ for all goals $g$. Thus, $\Delta V_{g_r}(s_1) = \Delta V_g(s_1)$, meaning the initial state is always deceptive, regardless of the value of $\alpha$.
>
> 3.  **Existence is Guaranteed:** Since any trajectory starts in a deceptive state ($s_1$) and must end in a truthful state (the real goal $g_r$), there must exist a "last" deceptive state along this path, after which all subsequent states are truthful.
>
> Therefore, while the computation becomes more complex, the core concept of the LDS as a critical transition point remains robust. The method for finding the optimal LDS, $d_c^\* = \arg\max_{s \in D} V_{g_r}(s)$, also remains applicable, where $V_{g_r}(s)$ and the deceptive set $D$ are now computed using the soft value functions.
>
> We do note, as discussed in [Savas et al., 2022], that if $\alpha$ is excessively large, the agent's perceived policy becomes almost random, and the notion of "deception" may lose its meaning. But for a wide range of sub-optimal yet goal-directed observers, our framework holds.

---

> ### Author Response · Authors · 2025-11-19
>
> **Q1: Why did you choose value differences (V-differences) instead of Q-differences as the foundation for deception modeling?**
>
> This is a key question that gets to the heart of our model design. We chose V-differences as our foundation primarily because a V-difference-based observer model aligns with the Markov property of DMDPs, leading to a more elegant and fundamental model.
>
> As we cite in our paper (lines 187-189), research by [Savas et al., 2022] notes that V-differences serve as a concise and effective approximation for plan recognition when transition randomness is low and the discount factor is high. In our DMDP setting, these conditions are met by definition.
>
> In contrast, using Q-differences would introduce a dependency on specific actions. The observer's belief would depend not just on the agent's current state, but also on the last action taken to arrive there. In a deterministic environment, where the state itself contains all necessary information for future decisions, considering the previous action is redundant. An observer is more concerned with "where you are now" than "how exactly you got here." This added complexity in the observer model is unnecessary.
>
> In summary, we chose V-differences because they provide a more fundamental, parsimonious, and consistent modeling basis for the DMDP problem structure. This choice enabled us to clearly define "deceptive" and "truthful" states, which in turn allowed us to build our theoretical framework around the **Last Deceptive State (LDS)** and achieve efficient two-phase deceptive planning.
>
> **Q2: How does your definition of deception align with human's intuitive notion of deception?**
>
> Our framework aligns with the intuitive, strategic nature of human deception on two core levels:
>
> 1.  **Operationalizing the Classic Definition of Deception:** As we note in our introduction, a classic definition of deception is "to cultivate and sustain a false belief in the mind of another" (Carson, 2010). Our VDM framework directly operationalizes this: **Cultivating and Sustaining  False Belief:** Stage 1 of our model (the Deceptive Phase), whether through ambiguity (A-VDM) or exaggeration (E-VDM), is explicitly designed to build a false model of the agent's intent in the observer's mind.
>
> 2.  **Capturing the Strategic Trade-off:** More deeply, our VDM framework captures a critical element of intelligent human deception: **deception is a means to an end, not the end itself.** A smart deceiver knows that deception has costs and that at some point, they must stop deceiving and turn to efficiently completing their true objective.
>
> **Intuitive Analogy:** In a military operation, a unit conducting a feint does not intend to feint forever. Its goal is to deceive the enemy up to a critical turning point (our LDS), after which it must rapidly pivot to the main objective. Endless, inefficient deception is not strategic and foolish.
>
> In conclusion, our work not only formally adheres to the classic definition of deception but also captures the intelligent, trade-off-aware nature of real-world strategic deception. We believe this consideration for efficiency makes our model a closer analogue to how humans employ deception strategically.
>
> We hope these responses have addressed your concerns. We’re happy to provide further clarification if needed. Thank you for your careful review—if our response has resolved your concerns, we would sincerely appreciate your consideration in updating the score.

---

> ### Author Response · Authors · 2025-11-28
>
> Dear Reviewer Soss,
>
> We sincerely hope this message finds you well. We would like to express our deepest gratitude for the time and effort you have dedicated to reviewing our submission and for providing such thoughtful and constructive feedback.
>
> We have carefully addressed all the questions and concerns raised in your reviews, and our detailed responses have been submitted through the rebuttal system. As the author-reviewer discussion period is approaching its conclusion (with less than one week remaining before the deadline), we wanted to kindly reach out to ensure that you have had the opportunity to review our responses.
>
> We understand that you have many responsibilities and reviewing demands on your time. However, if you have any remaining questions, concerns, or would like us to clarify any aspect of our rebuttal, we would be more than happy to provide additional information or elaboration. We are committed to engaging in a productive dialogue to address any outstanding issues.
>
> Your feedback has been invaluable in helping us improve our work, and we would greatly appreciate the opportunity to know whether our responses have adequately addressed your concerns. If our responses have resolved the issues you raised, we would be deeply grateful if you could kindly consider acknowledging this in the discussion thread or updating your assessment accordingly.
>
> Once again, thank you very much for your careful consideration of our work. We look forward to hearing from you at your earliest convenience.
>
> With sincere appreciation and respect,
>
> The Authors

---

### Author Response · Authors · 2025-11-19

# Revisions Made to ICLR Paper

## Detailed List of Modifications

1. **Line 158**: Changed "T" to $\infty$ in the value function definition.

2. **Line 209**: Added the clarifying statement:
   "We assume that all goal states are absorbing, i.e., $ P(g, a, g)  =  1$ for all  $g \in G$."

3. **Line 187**: Revised the text from:
   "The probability calculation of goal states based on V-differences exhibits a natural alignment with DMDPs"
   To:
   "The probability calculation based on V-differences is well-suited for DMDPs given their deterministic transitions."

4. **Line 270**: Modified the phrasing from "any action" to "there exists an action" for mathematical precision.

5. **Line 315**: Replaced the verb 'establish' with 'identify' for better academic phrasing.

6. **Line 317**: Enhanced the description of $g^\*$ from:
   "$g^\*$ possesses the strongest deceptive potential among all fake goal states at $d_c^\*$."
   To the more precise:
   "$g^\*$ represents the decoy goal state with the maximum posterior probability given the agent's state $d_c^\*$, making it the decoy goal state most likely to be mistaken for the true one."

7. **Chapter 6**:
   - Added detailed computer specifications for experimental reproducibility
   - Incorporated additional experimental results
   - Included metrics for deceptive planning generation time

---

### Comment · Area_Chair_DwX9 · 2025-11-29

Thank you for all the efforts in the rebuttal phase. Due to the ICLR new policy, I will take over all the leftovers. No need to worry about the unresponsive comments. Simply try your best to address the original reviews and apply them to your revised manuscript. I will check them carefully with a thorough discussion with SAC if necessary.

Plus, can you highlight (or change the color of) the section/paragraph that you have revised?

Your New AC

---

> ### Author Response · Authors · 2025-11-29
>
> Dear New AC,
>
> Thank you very much for your reply and for taking over our submission. As per your request, we have highlighted all revised sections in the manuscript using yellow color.
>
> During the rebuttal phase, we carefully studied all reviewers' comments and uploaded our revised manuscript along with detailed responses approximately ten days ago. We have made our best effort to address every question raised and have supplemented new experimental results based on the reviewers' suggestions.
>
> We particularly value the constructive feedback from Reviewer hQMd (who gave us a score of 2), and have provided detailed justifications and additional experiments in our response. Regarding some reviewers' comments that our approach is "simple," we would like to emphasize that simplicity, practical applicability, and generalizability are among the core strengths of our work.
>
> We sincerely hope to receive your evaluation and feedback on our revision efforts. Thank you again for your time and dedication to our paper during this extraordinary period!
>
> Best regards,
>
> Authors

---

### Meta-Review · Area_Chair_DwX9 · 2026-01-08

**Summary:**

The paper introduces a framework for strategic deception in Deterministic Markov Decision Processes (DMDPs) using the Last Deceptive State (LDS) and V-differences-based Deceptive Models (VDMs), aiming to improve deception efficiency. While the reviewers generally agree that the paper is sound and theoretically grounded (all reviewers), there are several concerns that influenced their evaluations.

While the theoretical contributions of the paper are generally well-regarded, Reviewer hQMd and Reviewer Soss raise concerns about the practical impact of the approach, particularly in real-world, stochastic environments with sub-optimal observers. The authors’ responses, including the extension to non-perfectly rational observers, were seen as insufficient in addressing these issues. As a result, the reviewers recommend marginally below the acceptance threshold, with the need for further clarification, comparisons, and experimental validation. The reviewers' primary concern centers around the real-world applicability and scalability of the proposed model.

**Reviewer Concerns:**

- Reviewer Soss and Reviewer qxpt both point out that the framework is limited to deterministic environments, and extending it to stochastic or partially observable environments is not addressed adequately. The authors justify this focus, but the reviewers express concerns about the real-world applicability of the model, especially in settings with uncertainty and non-ideal observers. Reviewer Soss highlights that the applicability of the findings to real-world scenarios is unclear.
- Reviewer Soss and Reviewer qxpt raise concerns about the assumption of a perfectly rational observer, which is unrealistic in many real-world settings. The authors do address this by introducing a "Boltzmann-rational" observer for non-perfect rationality, but Reviewer hQMd is skeptical that this fully resolves the practical relevance of the model in realistic contexts.
- Reviewer hQMd questions whether the LDS concept truly offers a significant improvement over the Last Deceptive Point (LDP) from prior work. The reviewer finds the distinction between LDS and LDP to be minor, particularly in deterministic settings. The authors respond by emphasizing the value-based approach of LDS, which allows for broader applicability, but this clarification may not fully address the reviewer’s concerns.
- Reviewer hQMd critiques the theoretical contributions, arguing that the formalization of LDS and its related concepts is not particularly novel. The reviewer suggests that the paper primarily introduces new definitions, which they feel do not provide a substantial theoretical advancement over previous works.
- Reviewer hQMd also points out the lack of comparison to other relevant methods, especially those proposed by Savas et al., 2022. The authors initially did not compare their approach to this important baseline, but after the review, they plan to include this comparison. The reviewer views this comparison as necessary to assess the novelty and effectiveness of the proposed framework.
- Reviewer qxpt asks for clarification on the computational complexity of LDS and whether the current method can scale to large state spaces. The authors acknowledge this limitation and mention potential future work using function approximation techniques like deep learning for larger state spaces. However, this limitation remains a concern for scalability.

**Reviewer Scores:**

There is no evidence that any reviewer will change their scores.

---

### Decision · Program_Chairs · 2026-01-26

Reject